# In Vitro Study of the Proliferation of MG63 Cells Cultured on Five Different Titanium Surfaces

**DOI:** 10.3390/ma17102208

**Published:** 2024-05-08

**Authors:** Roberto Campagna, Valentina Schiavoni, Enrico Marchetti, Eleonora Salvolini, Andrea Frontini, Francesco Sampalmieri, Fabrizio Bambini, Lucia Meme’

**Affiliations:** 1Department of Clinical Sciences, Polytechnic University of Marche, 60121 Ancona, Italy; r.campagna@univpm.it (R.C.); valentinaschiavoni03@gmail.com (V.S.); f.sampalmieri@univpm.it (F.S.); l.meme@staff.univpm.it (L.M.); 2Department of Life, Health, Environmental Sciences, University of L’Aquila, 67100 L’Aquila, Italy; enrico.marchetti@univaq.it; 3Department of Life and Environmental Sciences, Polytechnic University of Marche, 60121 Ancona, Italy; a.frontini@univpm.it

**Keywords:** MG63, implant surface, osteointegration, surface characteristics

## Abstract

The use of dental implants for prosthetic rehabilitation in dentistry is based on the concept of osteointegration. This concept enables the clinical stability of the implants and a total absence of inflammatory tissue between the implant surface and the bone tissue. For this reason, it is essential to understand the role of the titanium surface in promoting and maintaining or not maintaining contact between the bone matrix and the surface of the titanium implant. Materials and Methods: Five types of titanium discs placed in contact with osteoblast cultures of osteosarcomas were studied. The materials had different roughness. Scanning electron microscopy (SEM) photos were taken before the in vitro culture to analyze the surfaces, and at the end of the culturing time, the different gene expressions of a broad pattern of proteins were evaluated to analyze the osteoblast response, as indicated in the scientific literature. Results: It was demonstrated that the responses of the osteoblasts were different in the five cultures in contact with the five titanium discs with different surfaces; in particular, the response in the production of some proteins was statistically significant. Discussion: The key role of titanium surfaces underlines how it is still possible to carry out increasingly accurate and targeted studies in the search for new surfaces capable of stimulating a better osteoblastic response and the long-term maintenance of osteointegration.

## 1. Introduction

Titanium is the most widely used material in the creation of osseointegrated implants, which are devices used single or multiple times in dentistry for the replacement of teeth and for the treatment of edentulism. Different authors [1,2,3,4,5,6,7,8,9,10,11,12,13,14] observed that osseointegration is an implant–bone connection which does not feature intervening layers of soft tissue, although the connection is never 100%. Therefore, the problems of identifying the exact degree of bone attachment necessary for the implant to be considered osseointegrated have led to a definition of osseointegration based on clinical stability rather than on histological criteria: “A process thanks to which it is obtained, and maintains a rigid, clinically asymptomatic fixation of alloplastic material in the bone during functional loading” [15]. In fact, it has been demonstrated that in some cases, implants with a low amount of bone in contact with the implant were actually stable [16]. Dental implants have been made from different materials, including metals such as steel, gold alloys and titanium, and ceramics. However, according to numerous studies, the material that is most accepted by the body and therefore the most widely used is titanium. Titanium can be used in pure form or as an alloy. The most widely used alloy contains 6% aluminum and 4% vanadium (Ti6A4V). The scientific work we want to mention in which the properties of Ti6Al4V are studied is very interesting. This type of alloy, in addition to possessing biological characteristics, also has excellent mechanical resistance, which makes it particularly suitable for use in orthopedics and dentistry. In this study of Yingdi Yan et al., the topography and wettability of the surface play an important role in film formation and protein adhesion, following osseointegration, and even the duration of the inserted implant was investigated. In this paper, they prepared Ti-6Al-4V alloy samples using different smoothing and polishing materials as well the air plasma treatment, on which contact angles of water, formamide and diiodomethane were measured [17]. During the preparation processes of the system, the surface is made irregular through treatments with acids, sandblasting or plasma spray and is exposed to contact with the air, which will lead to a rapid formation of a layer of Ti oxide (TiO_2_); this negatively loads the plant, increasing its affinity for different biomolecules. The oxide also prevents the diffusion of metal ions within the tissues and therefore gives titanium a high degree of biocompatibility. If the oxide layer is removed during implant placement, it will reform within a millisecond. It is thanks to these characteristics that it is considered a bio-inert material, inducing the formation of bone in contact with the implant surface without the interposition of scar tissue [18,19].

It is important to remember that the chemical bonds between titanium and the organic matrix are weak bonds, and therefore the stability of the implant is essentially due to a purely mechanical retention; this interaction is therefore conditioned by the topography of the titanium surface. In fact, the rough surfaces give the material greater resistance to the tensile and torsional forces that develop at the interface, thanks to the greater surface area available for bone contact [20,21]. Although the bone tissue, complete with cells and fundamental substances, needs a minimum thickness of 100 µm to grow within the asperities, the mineral component adapts to surface irregularities of variable dimensions between 1 and 100 µm, explaining why modifications to the surface topography at this level have a profound impact on the holding power of the implant [22,23]. Over the years it has been possible to observe in increasing detail the way in which the organism responds to the insertion of osseointegrated implants within the bone tissue. In particular, Sela et al. [24] described how the response is mediated directly and indirectly; in both modalities, we witness the direct intervention of the growth factors contained within the platelets, which immediately intervene at the time of surgery. The role of TGFbeta in chemotaxis towards undifferentiated osteoblasts and of PDGF, platelet-derived growth factor, which intervenes in neoangiogenesis, essential for the neovascularisation of the clot, has been well described. This entire process is mediated directly and indirectly by hormonal factors. The result consists of the release of the vesicle matrix, which initiates the calcification front and therefore the deposition of a new bone matrix. This physiological healing process intervenes in bone repair, in the case of osseointegrated implants, and ends up confronting the implant surfaces. Titanium surfaces have therefore been subjected over the years to different types of treatment to improve their responses in terms of osteoblastic deposition. Hydroxyapatite-coated implants have also been used to increase the deposition of the bone matrix [25]. The characteristics of roughness and biological affinity peculiar to this type of implant have been proven for a long time. The role of the type of implant surface is therefore essential in trying to obtain a greater quantity of bone matrix, and therefore greater mechanical anchoring and consequently greater performance in the role that integrated implants must play, i.e., the role of supporting dental prostheses of different types. Many authors have tried to modify the surface of titanium by adding substances that could induce greater bone deposition. It is worth mentioning the study conducted by Meme et al. [26,27,28,29], in which Raloxifene was combined with titanium in a pre-silanization process. Raloxifene is a molecule capable of directly regulating the differentiation of osteoblasts according to a mechanism enacted by SERMs (Selective Estrogen Receptor Modulators). Particular mention should be made of the work conducted by Orsini et al. [30] for the creation of a titanium surface, later patented with the name of Nanotite, to which nanoparticles of calcium sulphate were attached; this surface reduced the physiological resorption curve observed during the first 8 weeks by several weeks. Another type of surface capable of improving the osteoblastic response is the one produced by a well-known implant company and patented under the name of Roxolid (Straumann, Basel, Switzerland) [30], on which there are layers of titanium peroxide obtained by keeping the implants in ampoules with hydrogen peroxide. However, in the literature we also find scientific works that demonstrate [31] how hooking something onto the implant surfaces could expose the implant to failure due to the detachment of said materials; even the response of the oral soft tissues, epithelium and connective tissue does not produce a totally positive response, and implant diseases (like implantitis) triggered by the roughness of the surfaces have been described. These types of results have led some researchers, in particular Simion et al. [32,33], to go back to machined implants, without treatment. In the literature, there have been some interesting works on the improvement of the osteoblastic response obtained without altering the bone surface, but only by using particular cap screws for implants equipped with static magnetic fields [34,35,36,37]. The static magnetic fields according to these studies have been demonstrated, both in vitro and in vivo, to increase osteoblast differentiation and increase bone matrix deposition. This type of application avoids the release into the tissues of materials previously subjected to adhesion and subsequently detached from the surface by not altering the surface of the titanium.

To date, therefore, there is no titanium surface that can be considered the absolute best in terms of osteoblastic and epithelial response, thus investigating the effects of the different titanium surface on cell proliferation and osteogenic gene expression could contribute to clarify which titanium surface could be considered the best material.

The aim of this work was to study in vitro the different responses of osteoblasts cultured on five different types of titanium surfaces. In particular, 5 surfaces were compared, in particular one of these, disc 4, which is completely new: disc no. 1 did not undergo any treatment, disc no. 2 was polished by means of electroerosion, disc no. 3 was sandblasted and double-acid-etched, the surface of disc no. 4 was a new Al Ti Color surface, and disc no. 5 was subjected to color anodizing. Cells exposure to different types of titanium surfaces was able to affect cell proliferation, and specific surface treatments demonstrated to potentiate the ability of osteogenic cells to proliferate. Moreover, sandblasted and double-acid-etched surfaces were able to affect gene expression, boosting the expression of main genes involved in osteogenesis.

## 2. Materials and Methods

### 2.1. Experimental Design

In the present study, MG63 cells were cultured both directly and indirectly in contact with discs made of Ti6Al4V grade 5, 10 mm in diameter. Discs were manufactured by New Ancorvis S.r.l. (Bargellino di Calderara di Reno, Italy) and subjected to different surface treatments performed by Al Ti Color S.r.l, (Piazzola sul Brenta, Italy). Specifically, disc no. 1 did not undergo any treatment, disc no. 2 was polished with electroerosion, disc no. 3 was subjected to sandblasting and double acid etching, the surface of disc no. 4 was a new Al Ti Color surface, and disc no. 5 underwent color anodizing (Figure 1). Investigators were made aware of the surface treatment only at the end of the study. Scanning electron microscopy (SEM) was performed for the instrumented implants in order to evaluate any type of modifications. The samples were cleaned with water spray, and surface topography was evaluated using SEM (Carl Zeiss Gemini SEM 500; Carl Zeiss, Oberkochen, Germany) operating at 10 kV with a working distance of 9 mm.

### 2.2. Cell Cultures

The human osteosarcoma cell line MG63, a well-established cell model, was obtained from the American Type Culture Collection (ATCC, Rockville, MD, USA). MG63 osteoblast-like cells were cultured and maintained in a monolayer in T25 cm^2^ culture flasks in High Glucose Dulbecco’s Modified Eagle’s Medium (DMEM High Glucose, Euroclone, Milan, Italy) supplemented with 10% fetal calf serum (FCS) and 1X Penicillin-Streptomicin (Euroclone), at 37 °C in a humified 5% CO_2_ incubator.

### 2.3. Cell Proliferation Assay

Cell proliferation was evaluated using a colorimetric assay that quantified the conversion of 3-(4,5-dimethylthiazol-2-yl)-2,5-diphenyl tetrazolium bromide (MTT, Merk Life Science S.r.l., Milan, Italy) to insoluble formazan through dehydrogenase enzymes of the undamaged mitochondria of living cells, as previously described [36]. Briefly, MG63 proliferation was assessed at different timepoints (0, 24, 48, and 72 h) in the presence of titanium discs subjected to different surface treatments. Discs with untreated surfaces served as controls.

The measurements were carried out on cells grown on the disc surface as well as on those indirectly in contact with discs. The indirect contact was achieved by placing discs in transwells (Greiner Bio-One S.r.l., Cassina de Pecchi, Italy) and allowing the exchange of the medium between the upper chamber and the lower chamber containing cells.

As concerns MG63 cultured on the disc surface, 3 × 10^4^ cells were seeded onto titanium discs in 24-well plates and allowed to attach for 5 h in the incubator. Subsequently, the medium was removed and replaced with 700 µL of fresh DMEM High Glucose. For each timepoint, cell proliferation was analyzed by measuring the conversion of tetrazolium salt MTT to formazan crystals. For this purpose, 700 μL of complete fresh medium containing 58.3 µL of MTT reagent (5 mg/mL in phosphate-buffered saline) was added to each well. After 2 h of incubation at 37 °C, the solution was discarded and the formazan crystals were dissolved by adding 200 μL of 2-propanol (Merck, Darmstadt, Germany) to each well. They were then transferred to a 96-well plate for absorbance measurements.

Regarding the indirect contact test, 3 × 10^4^ MG63 cells were seeded in 24-well plates with complete culture medium and allowed to attach overnight. The day after, discs were placed in permeable 8 µm 24-well transwells, allowing the exchange of medium between the upper chamber and the lower chamber containing cells. Then, 41.6 μL of MTT reagent (5 mg/mL in phosphate-buffered saline) dissolved in complete fresh medium (500 μL/well) was added to each well. Following an incubation of 2 h at 37 °C, the solution was removed and 200 μL of 2-propanol was added to each well to dissolve the formazan crystals, as previously described [37].

The reaction product was quantified by assessing the absorbance at 540 nm using a microplate reader (Multiskan GO, Thermo Fisher Scientific Inc., Waltham, MA, USA). Results were expressed as a percentage of the control (control equals 100% and corresponds to the absorbance value of each sample at time 0) and presented as mean values ± standard deviation of three independent experiments performed in triplicate.

### 2.4. Fluorescence Microscopy

The adhesion of cells to the discs was evaluated by means of fluorescence microscopy. First, 3 × 10^4^ cells were seeded onto titanium discs in 24-well plates, as described above. After 72 h, the medium was removed and cells were washed once with 700 µL 1X PBS (potassium chloride 2.7 mM; potassium phosphate monobasic 1.76 mM; sodium chloride 0.137 M; sodium phosphate dibasic 10.1 mM; pH 7.4) and fixed with 4% formaldehyde for 15 min at room temperature. Subsequently, the samples were washed three times with 1X PBS and then stored at 4 °C in 700 µL of 1X PBS until the time of analysis.

Then, 4′,6-diamidino-2-phenylindole (DAPI, Prodotti Gianni S.r.l., Milan, Italy), a blue fluorescent DNA stain that is commonly used as a nuclear counterstain in fluorescence microscopy, was used to detect cells on discs. First, 100 µL of mounting medium with DAPI was placed on microscope slides (Bio-Optica, Milan, Italy) and discs were positioned so that the surface with the attached cells was in contact with the DAPI. After an incubation of 10 min at room temperature, slides were observed by means of a NIKON AIR confocal fluorescence inverted microscope (Nikon Corporation, Tokyo, Japan) with a 20X objective, and the images were acquired using NIS-Element imaging and analysis software (version 5.21.00; Nikon).

### 2.5. RNA Extraction and Reverse Transcription

MG63 cells were trypsinized with 500 µL of trypsin-EDTA 1X (Euroclone, Milan, Italy) for 5 min at 37 °C. Trypsin was neutralized by 1 mL of full medium and cells were centrifuged at 500× *g* for 5 min. Cell pellet was resuspended in 1 mL of PBS 1X and recentrifuged. Upon last centrifugation, the supernatants were subsequently discarded, and the resulting cell pellets (3 × 10^5^) were homogenized. Total RNA was isolated using the SV Total RNA Isolation System (Promega, Madison, WI, USA), according to the manufacturer’s protocol. After estimating RNA purity and quantity by nanodrop, 1 µg of total RNA was reverse-transcribed with random primers in a total volume of 25 µL for 60 min at 37 °C, by means of the M-MLV Reverse Transcriptase kit (Promega), according to the manufacturer’s instructions, and the cDNA obtained was stored at −20 °C for further analyses.

### 2.6. Real-Time PCR

The human osteogenesis PCR array SBHS-026ZD supplied by Qiagen (Germantown, MD, USA) was used to profile the expression of 84 genes related to osteogenic differentiation at different timepoints.

The cDNA was mixed with QuantiNova™ SYBR^®^ Green PCR kit (Qiagen), and 20 μL aliquots were loaded into each well of the human osteogenesis PCR array. PCR array experiments were performed using the CFX96 Real-Time PCR Detection System (Bio-Rad Laboratories, Hercules, CA, USA). Conditions for amplification were as follows: 1 cycle of 2 min at 95 °C followed by 40 cycles of 5 s at 95 °C and 10 s at 60 °C.

The PCR array data were analyzed using the ΔΔCt method, as previously described [38,39,40]. Genes with Ct values ≥ 35 were considered not detectable (negative call) and assigned a value of 35. β-actin (ACTB) was used as a housekeeping gene to obtain the ΔCt value for each gene of interest. The fluorescence produced by the intercalating fluorescent dye, which binds to double strand DNA after every cycle, was utilized to monitor the direct detection of PCR product increase. Each gene examined was expressed as ΔCt value, where ΔCt = Ct (Gene of interest) − Ct (β-actin). Fold changes in relative gene expression were evaluated by 2^−ΔΔCt^ method, calculating ΔCt = Ct (Gene of interest) − Ct (β-actin) and Δ(ΔCt) = ΔCt (disc no. 1) − ΔCt (disc no. 2; disc no. 3; disc no. 4; disc no. 5).

### 2.7. Statistical Analysis

Results were analyzed using GraphPad Prism software 8.4.2 (GraphPad Software Inc., San Diego, CA, USA). Differences between groups were determined by a One-Way ANOVA test. A *p*-value < 0.05 was considered statistically significant.

## 3. Results

### 3.1. Scanning Electron Microscopy (SEM) Evaluation of Disc Surfaces

Scanning electron microscopy (SEM) was utilized to visualize the disc surfaces upon different surface treatment. Specifically, disc no. 1 did not undergo any treatment (Figure 1A), disc no. 2 was polished by means of electroerosion (Figure 1B), disc no. 3 was sandblasted and double-acid-etched (Figure 1C), the surface of disc no. 4 was a new Al Ti Color surface (Figure 1D), and disc no. 5 was subjected to color anodizing (Figure 1E). The surface of disc no. 1 appeared jagged due to the lack of a surface treatment following the alloy production. Disc no. 2 showed, instead, a porous surface, while the surface of disc no. 3 was characterized by the presence of depressions of different shape and size. Disc no. 4 showed a more irregular surface topography, with an alternation of smooth and rough areas. Finally, disc no. 5 exhibited a surface with intermediate characteristics between disc no. 1 and the others.

### 3.2. Cell Proliferation Assay

The proliferation of MG63 cells cultured in both direct and indirect contact with different discs was evaluated using the MTT assay at different timepoints.

Data concerning the direct contact test are shown in Figure 2.

After 72 h, cells grown on disc no. 2 showed a significantly lower proliferation rate than those seeded on the control disc, with results of 164.81 ± 16.51% for MG63 cultured in direct contact with the control disc and 130.40 ± 15.51% for cells grown on disc no. 2. Although the difference at 48 h was not statistically significant, a recovery of cell proliferation with respect to the control was observed (Figure 2A).

The contact with disc no. 3 induced a significant increase in cell proliferation compared to the control group at all the timepoints. In particular, at 24 h the proliferation of MG63 cells seeded on disc no. 3 was 249.00 ± 29.61%, while the control group exhibited a proliferation value of 90.84 ± 10.74%; this difference in proliferation, although statistically significant, showed a reduction at 48 h (131.78 ± 19.62% for cells cultured on the control disc and 235.46 ± 27.54% for cells grown on disc no. 3), and increased again after 72 h, when cells seeded on disc no. 3 reached a proliferation value of 380.08 ± 40.27% (Figure 2B).

As concerns MG63 cells cultured in direct contact with disc no. 4, a significant higher proliferation rate was recorded than in the control, starting from the 48 h-timepoint: 203.61 ± 23.12% at 48 h and 231.17 ± 24.74% at 72 h (Figure 2C).

Conversely, disc no. 5 triggered a significant reduction in cell proliferation at 72 h, when a value of 114.20 ± 16.38% was recorded (Figure 2D).

The results of the proliferation of MG63 cells cultured with indirect contact with discs are reported in Figure 3.

Cells exposed to disc no. 2 showed a decreased proliferation rate compared to those in indirect contact with the control disc at all the timepoints, although statistical significance was reached only at 48 h, with results of 157.10 ± 18.32% for cells in indirect contact with disc no. 2 and 254.36 ± 29.67% for MG63 exposed to the control disc. However, at the 72 h-timepoint, a tendency towards a lower proliferation gap was observed (Figure 3A).

Conversely, at 72 h, cells grown in indirect contact with disc no. 3 exhibited a proliferation of 383.47 ± 37.86%, which was significantly higher than that observed in MG63 exposed to the control disc (282.14 ± 29.31%) (Figure 3B).

Exposure to disc no. 4 induced a statistically significant increase in cell proliferation compared to the control group, starting 48 h after seeding: 331.91 ± 29.13% at 48 h and 354.61 ± 31.20% at 72 h (Figure 3C).

Similar to the data obtained for cells grown on the disc surface, the indirect contact with disc no. 5 led to a significant decrease in cell proliferation with respect to the control group at 72 h, when a value of 210.35 ± 24.37% was reached (Figure 3D).

### 3.3. Fluorescence Microscopy

In order to assess the ability of MG63 cells to properly adhere on different discs surfaces and the influence of these different surfaces on proliferation rate, cells were seeded on discs and 72 h after seeding were stained and imaged by fluorescence microscopy.

In line with the data previously obtained through the MTT assay, fluorescence microscopy demonstrated that when MG63 cells were seeded on discs no. 3 (Figure 4C) and 4 (Figure 4D), their proliferation rate was enhanced since a greater number of cells than the control (Figure 4A) was detected, as evidenced by nuclear staining with DAPI, while a lower cell density was observed for discs no. 2 (Figure 4B) and 5 (Figure 4E). The apparent higher fluorescence observed for the disc no. 5 compared to control is due to a higher background signal arising from its surface. Indeed, the irregular-rounded highly fluorescent circles are not nuclei, and the related fluorescence is due to the surface treatment of this disc.

### 3.4. Gene Expression Profiling

Variations in the expression of 84 genes involved in osteogenesis were examined. Among these 84 genes, 39 were selected as they were notably dysregulated in cells cultured on discs which underwent different surface treatment compared to those grown on the control disc. The expression of selected genes is reported in Figure 5, while Table 1 lists their fold expressions at all the timepoints analyzed.

In particular, our data showed a remarkable upregulation of osteogenesis-related genes, especially in cells grown on disc no. 3, although their expression was also increased in MG63 cells in contact with discs subjected to other surface treatments.

## 4. Discussion

Our data showed that sandblasting and double-acid etching, as well as the new Al Ti Color surfaces do not hinder MG63 cell adhesion. Moreover, the obtained results suggested a stimulating effect of these surface treatments on osteoblast-like cell proliferation. In addition, an evident upregulation of 39 genes involved in osteogenesis and bone turnover was observed, especially in cells grown on discs subjected to sandblasting and double acid etching. In particular, COL14A1, COL3A1, and SERPINH1 are related to collagen biosynthesis; CD36, CDH11, FN1, ICAM1, ITGA3, ITGB1, and VCAM encode cell adhesion molecules; and RUNX2, TGFB1, TGFBR1, TGFBR2, SMAD2, SMAD3, and TWIST1 are responsible for the differentiation of osteoblast precursor cells and the production of osteoid. Since the deposition of new bone implies physiological bone remodeling, the enhanced expression of CSF1, CTSK, MMP2, and NOG genes provides evidence of an active and robust coordinated process in which osteogenesis is coupled with bone remodeling.

Cells cultured on sandblasted and double-acid-etched discs also showed a remarkable upregulation of angiogenesis-related genes, including VCAM, VEGFA, and VEGFB; VEGF, in particular, is produced by osteoblasts during the early steps of bone regeneration and it plays a key role in blood vessels’ neoformation, crucially contributing to the complex process of bone deposition and remodeling. Lastly, osteoblast-like cells exposed to discs subjected to sandblasting and double-acid etching exhibited a notable overexpression of genes related to cell proliferation, such as ANXA5, BGN, BMP1, BMP2, BPM4, BMP6, EGF, EGFR, and GLI1, thus corroborating MTT assay results.

## 5. Conclusions

The search for surfaces that can prove more active in terms of the osteoblast response in terms of bone matrix production is still a very current objective; it is a real open challenge and many researchers have worked and are working towards this goal. The results in terms of a positive effect on the expression of key genes involved in osteogenesis, a crucial process in order to achieve optimal implant osseointegration, of our scientific work showed that even today the surface of the disc no. 3, sandblasted and double-acid-etched, is the one capable of obtaining the best results to date; the new surface of disc no. 4, which is a new Al Ti Color surface, gave excellent results at both 48 and 72 h. For this new surface we believe that in the future in vivo studies will need to be performed to evaluate the short and long term osteoblastic response precisely because the in vitro results have been very encouraging. In conclusion, our in vitro study highlighted that MG63 cell exposure to both the Ti6Al4V grade 5 subjected to sandblasting and double-acid etching and the new Al Ti Color surface can exert a positive boost to cell proliferation. The use of such surface treatments could therefore be successfully applied in the field of dental implantology. This study shows that it could be useful in treating the titanium and for the subtraction process, but not for the addition process with very good bone reaction without the risk of losing the materials attached to the titanium surfaces.

## Figures and Tables

**Figure 1 materials-17-02208-f001:**
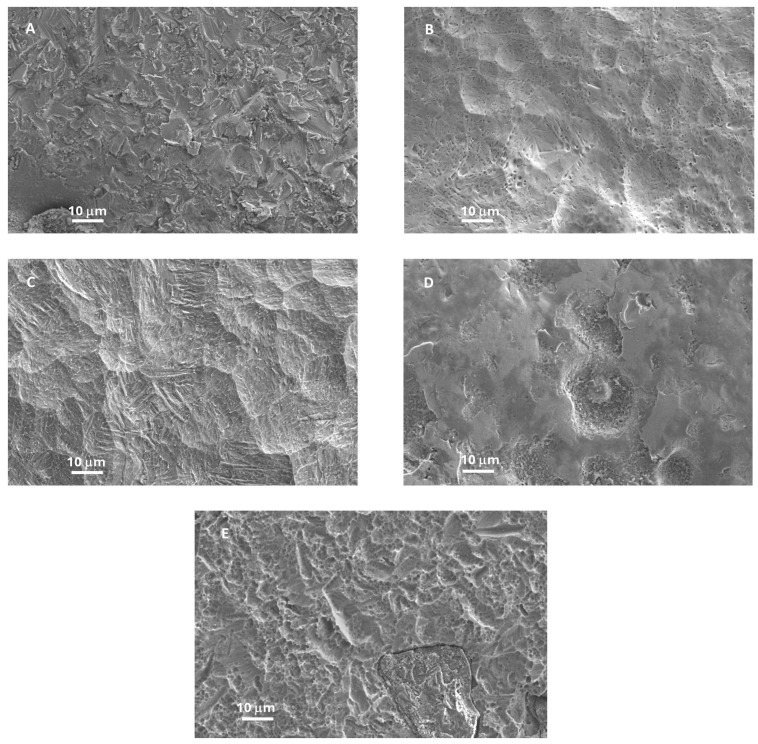
Scanning electron microscopy (SEM) evaluation of disc surfaces. (**A**): disc 1; (**B**): disc 2; (**C**): disc 3; (**D**): disc 4; (**E**): disc 5. All pictures were taken at ×1000 magnification.

**Figure 2 materials-17-02208-f002:**
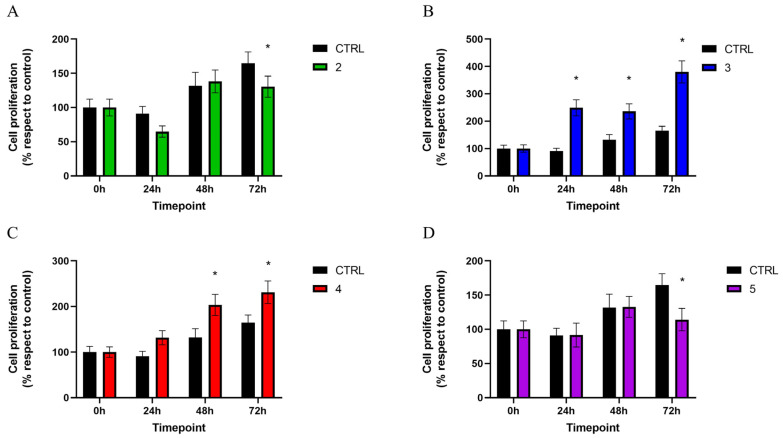
MTT assay was performed at different timepoints (0, 24, 48, and 72 h) to evaluate the proliferation of MG63 cells cultured on discs 2 (**A**), 3 (**B**), 4 (**C**), and 5 (**D**) with respect to those grown on control disc (CTRL). Values are expressed as mean ± standard deviation; * *p* < 0.05.

**Figure 3 materials-17-02208-f003:**
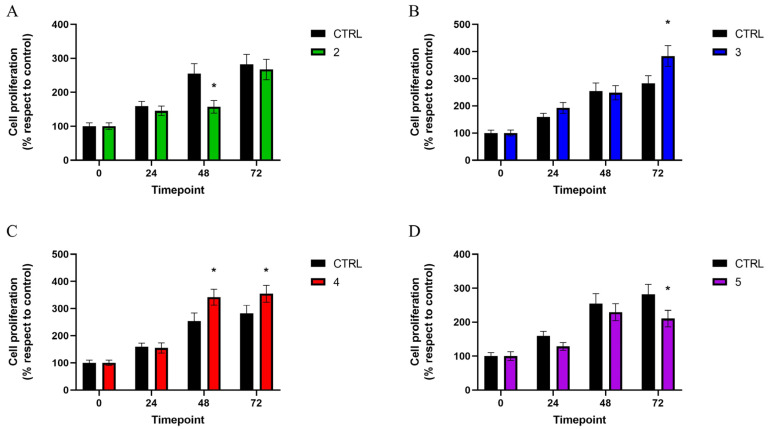
MTT assay performed at different timepoints (0, 24, 48, and 72 h) to evaluate the proliferation of MG63 cells cultured on discs 2 (**A**), 3 (**B**), 4 (**C**), and 5 (**D**) with respect to those grown on the control disc (CTRL). Values are expressed as mean ± standard deviation; * *p* < 0.05.

**Figure 4 materials-17-02208-f004:**
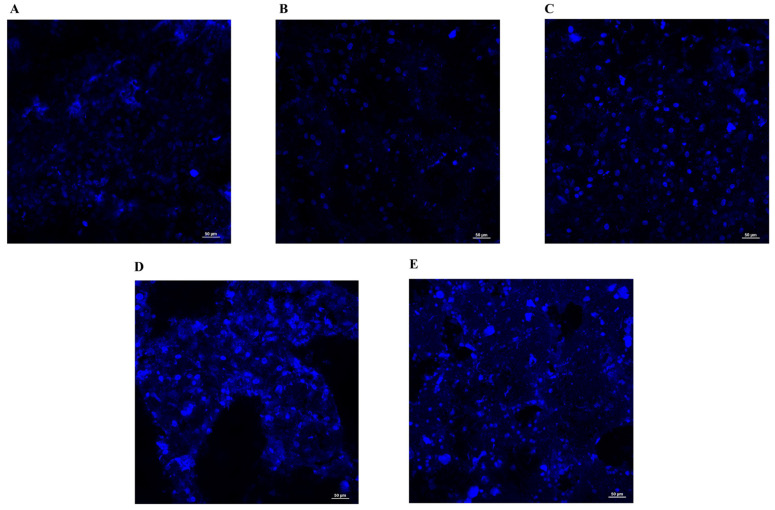
Fluorescence microscopy images of viable MG63 cells cultured on control disc (**A**), disc no. 2 (**B**), disc no. 3 (**C**), disc no. 4 (**D**), and disc no. 5 (**E**) at 72 h. DAPI was used for nuclei staining.

**Figure 5 materials-17-02208-f005:**
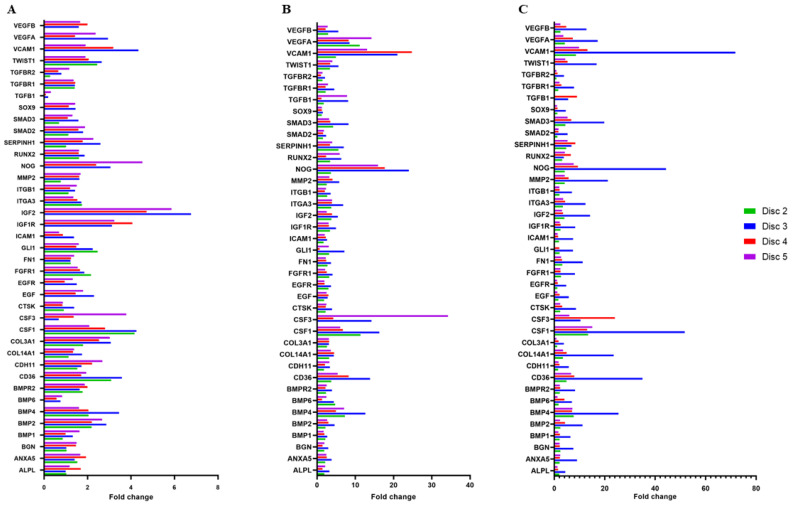
Differential expression of 39 genes dysregulated in MG63 cells cultured on discs no. 2, 3, 4, and 5 with respect to the control at 24 (**A**), 48 (**B**), and 72 h (**C**).

**Table 1 materials-17-02208-t001:** Fold expression of 39 dysregulated genes in MG63 cells cultured on discs no. 2, 3, 4, and 5 with respect to the control at 24, 48, and 72 h.

	**Timepoint 24 h**			
**Gene**	**Disc 2**	**Disc 3**	**Disc 4**	**Disc 5**
ALPL	1.02	0.99	1.69	1.18
ANXA5	1.53	1.40	1.93	1.66
BGN	1.03	1.02	1.47	1.49
BMP1	0.85	1.32	0.98	1.63
BMP2	2.19	2.86	2.19	2.67
BMP4	2.05	3.45	2.04	1.61
BMP6	Not detectable	0.75	0.57	0.83
BMPR2	1.77	1.64	2.00	1.87
CD36	3.09	3.58	1.71	1.94
CDH11	1.53	1.72	2.21	2.68
COL14A1	1.12	1.74	1.33	1.38
COL3A1	1.79	3.07	2.52	3.02
CSF1	4.17	4.25	2.80	2.08
CSF3	0.02	0.67	1.36	3.78
CTSK	0.90	1.38	0.84	0.86
EGF	Not detectable	2.29	1.45	1.79
EGFR	Not detectable	1.50	0.95	1.31
FGFR1	2.16	1.84	1.65	1.55
FN1	1.22	1.21	1.24	1.38
GLI1	2.46	2.24	1.48	1.59
ICAM1	Not detectable	1.37	0.87	0.68
IGF1R	Not detectable	3.13	4.06	3.23
IGF2	Not detectable	6.76	4.72	5.86
ITGA3	1.74	1.71	1.53	1.34
ITGB1	1.13	1.42	1.20	1.50
MMP2	0.76	1.62	1.61	1.68
NOG	Not detectable	3.06	2.39	4.53
RUNX2	1.61	1.86	1.60	1.60
SERPINH1	1.01	2.59	1.77	2.27
SMAD2	1.12	1.79	1.58	1.88
SMAD3	0.69	1.58	1.09	1.30
SOX9	Not detectable	1.44	1.14	1.43
TGFB1	0.00	0.18	0.08	0.31
TGFBR1	1.41	1.42	1.43	1.36
TGFBR2	0.28	0.79	0.65	1.16
TWIST1	2.44	2.65	2.05	1.91
VCAM1	Not detectable	4.34	3.19	1.91
VEGFA	Not detectable	2.93	1.42	2.37
VEGFB	Not detectable	1.59	1.99	1.66
	**Timepoint 48 h**			
**Gene**	**Disc 2**	**Disc 3**	**Disc 4**	**Disc 5**
ALPL	1.92	3.22	1.41	2.12
ANXA5	2.13	3.79	2.55	2.46
BGN	1.87	2.96	1.45	1.91
BMP1	2.17	2.66	1.92	1.72
BMP2	2.21	4.60	2.99	2.66
BMP4	7.26	12.63	4.96	7.08
BMP6	4.73	4.39	1.31	2.47
BMPR2	2.31	3.87	2.23	2.53
CD36	3.72	13.85	8.25	5.40
CDH11	1.79	3.37	2.09	3.24
COL14A1	3.13	4.39	4.44	3.61
COL3A1	2.62	3.11	3.15	3.11
CSF1	11.38	16.28	6.75	6.04
CSF3	0.11	14.22	4.24	34.25
CTSK	2.26	3.94	2.31	2.44
EGF	1.84	2.79	2.98	2.50
EGFR	2.98	3.64	1.96	1.94
FGFR1	3.18	4.06	2.52	2.15
FN1	2.73	3.63	2.40	2.31
GLI1	3.13	7.18	0.71	3.07
ICAM1	1.72	2.63	2.30	1.99
IGF1R	3.38	4.95	3.11	3.03
IGF2	3.76	5.43	3.91	2.54
ITGA3	3.72	6.83	3.87	3.94
ITGB1	2.66	3.56	2.06	2.32
MMP2	2.53	5.79	4.03	3.23
NOG	3.65	24.03	17.71	15.95
RUNX2	3.44	6.33	2.37	5.93
SERPINH1	5.62	7.00	3.40	3.96
SMAD2	1.56	2.38	1.61	1.78
SMAD3	4.23	8.22	3.47	3.14
SOX9	1.24	1.50	1.25	1.25
TGFB1	1.78	8.13	1.16	7.84
TGFBR1	2.25	4.51	2.27	2.88
TGFBR2	1.45	2.09	1.03	1.44
TWIST1	3.38	5.61	3.46	4.00
VCAM1	4.96	21.01	24.79	13.08
VEGFA	11.15	8.53	8.24	14.25
VEGFB	2.91	5.56	2.25	2.80
	**Timepoint 72 h**			
**Gene**	**Disc 2**	**Disc 3**	**Disc 4**	**Disc 5**
ALPL	1.97	4.38	1.53	1.25
ANXA5	2.12	9.00	2.25	2.32
BGN	2.39	7.56	2.19	2.08
BMP1	2.09	6.35	2.24	1.63
BMP2	2.39	11.17	4.26	2.31
BMP4	7.67	25.46	7.04	7.17
BMP6	1.67	6.99	4.03	1.28
BMPR2	2.16	8.20	2.33	2.14
CD36	4.82	34.95	7.94	6.69
CDH11	1.54	5.76	2.07	1.67
COL14A1	3.39	23.55	4.88	3.46
COL3A1	1.14	3.75	1.67	0.84
CSF1	13.39	51.72	13.04	15.06
CSF3	Not detectable	10.39	23.99	5.94
CTSK	2.32	8.57	3.13	2.34
EGF	1.63	5.73	2.12	1.23
EGFR	1.21	4.72	1.38	1.07
FGFR1	2.62	8.17	2.46	2.43
FN1	3.18	11.26	3.06	2.70
GLI1	2.09	7.41	2.00	Not detectable
ICAM1	1.88	7.48	1.51	1.35
IGF1R	2.48	8.31	2.50	2.14
IGF2	4.01	14.18	3.48	3.05
ITGA3	3.39	12.36	4.29	3.53
ITGB1	2.05	6.95	2.12	1.99
MMP2	4.14	21.23	5.69	4.11
NOG	4.15	44.30	9.35	7.69
RUNX2	3.12	3.77	6.59	4.20
SERPINH1	4.70	6.81	8.32	5.27
SMAD2	1.20	5.25	1.65	1.68
SMAD3	4.40	19.80	6.73	5.27
SOX9	1.21	4.54	1.29	1.13
TGFB1	0.08	5.46	8.97	0.20
TGFBR1	1.58	7.82	2.82	2.03
TGFBR2	0.88	3.85	1.29	0.87
TWIST1	Not detectable	16.77	5.24	4.37
VCAM1	8.60	71.78	13.14	9.82
VEGFA	4.16	17.19	7.34	3.60
VEGFB	2.44	12.75	4.75	2.42

## Data Availability

Data is contained within the article.

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
