# Peer review of "In Vitro Study of the Proliferation of MG63 Cells Cultured on Five Different Titanium Surfaces"

_materials, 2024, doi:10.3390/ma17102208_

Round 1

Reviewer 1 Report

Comments and Suggestions for Authors

Manuscript "In vitro and SEM study of the proliferation of MG63 cells cultured on five different titanium surfaces", by Lucia Meme, Roberto Campagna, Enrico Marchetti, Valentina Schiavoni, Andrea Frontini, Francesco Sampalmieri, Fabrizio Bambini and Eleonora Salvolini.

The research of a contemporary problem is presented in the manuscript. The methods for analysis are well chosen, the results are significant. Small corrections in the manuscript are also needed.

The manuscript states: "while a lower cell density was observed for discs no. 2 (Figure 4B) and 5 (Figure 4E)." (lines 310 and 311). In Figure 4E for disc no. 5 shows a higher cell density than the control disc. I ask the authors to clarify their statement.

I consider that the paper should be published in the journal Materials after minor corrections.

Author Response

The manuscript states: "while a lower cell density was observed for discs no. 2 (Figure 4B) and 5 (Figure 4E)." (lines 310 and 311). In Figure 4E for disc no. 5 shows a higher cell density than the control disc. I ask the authors to clarify their statement.

We thank the reviewer for underlying this point. The apparent higher fluorescence observed for the disk 5 compared to control is due to a higher background signal arising from the surface of disk 5. Indeed, the irregular-rounded highly fluorescent circles are not nuclei, and the related fluorescence is due to the surface treatment of this disk. However, we agree that this information should be specified in the text of the manuscript in order to avoid misunderstandings with the readers, and thus the paragraph 3.2 of the manuscript has been modified accordingly.

Reviewer 2 Report

Comments and Suggestions for Authors

Overall, the work is well-structured, and this contribution should be considered for publication after addressing the following comments.

1.     Modify your resume, write the complete name of SEM as well provide some numerical data.

2.     The introduction is ok in general, informative, and with an adequate amount of details. Still, the novelty of the investigation and the anticipated results should be clearly stated.

3.     In figure 1 provide the scale Bar

4.     Section 2.4 Why the samples were washed with PBS(how much concentration) write the detailed procedure

5.     In Section 2.5 cell  were trypsinized, centrifuged (write in detail rpm, how much time

6.     Line 232  ΔΔCt method write in deital

7.     Section Fluorescence microscopy line 307-t0 line 311 is confusing please rewrite.

8.     The conclusion is too simple it can be improved further.

Reviewer 3 Report

Comments and Suggestions for Authors

This paper presents experimental results of in vitro studies on the proliferation of MG63 cells cultured on the two-phase TiAl6V4 alloy with the commercial name Protasul 64WF, which was initially developed for aviation industry requirements. Despite the material demonstrating rather good susceptibility to plastic deformation, its high hardness and rigidness compared to bones is a significant flaw. Furthermore, due to the possibility of damaging the surface of elements manufactured using this type of alloy as a result of abrasive and corrosive wear, there are justifiable fears concerning the harmful influence of aluminium and vanadium on the human body, which can lead to neurological disorders, such as Alzheimer’s or Parkinson’s disease. The authors took into consideration in their investigations the influence of the TiAl6V4 alloy surface preparation method. On the basis of appropriately adopted research methodology, the authors achieved the goal of their studies. The overall quality of this experimental work and writing is good. The results obtained from the investigations might be interesting for researchers working in similar fields. The manuscript can be recommended for publication after a few amendments which should be made in order to make the paper more accessible to readers.

1. Introduction, page 1, line 34: Albrektsson and Zarb are not the only authors of cited works [1-14].

2. Introduction: information is missing in the section of the work on the fundamental physicochemical properties of TiAl6V4 alloy, which is the most well-known material used in orthopedic and dental implantology.

3. Introduction, page 3, lines 114-115: the goal of the work is not very precisely formulated in the context of widely available literature on the subject of TiAl6V4 alloy biological studies.

4. Experimental design, page 3: Figure 1 should be moved to the Results section and SEM morphological observations of the investigated sample microstructures - discussed in detail. Moreover, morphological observations of the samples after biological studies are not included in this paper.

5. Investigations on titanium alloys up to now indicate that the aphase dissolves large amounts of oxygen. Dissolved oxygen influences the mechanical properties of titanium, e.g., it increases its hardness and brittleness. Do the authors believe that this will have an effect on the biomechanical properties of the biocomposites obtained in the work?
